# The Role of Root Exudates of Barley Colonized by *Pseudomonas fluorescens* in Enhancing Root Colonization by *Fusarium culmorum*

**DOI:** 10.3390/plants9030366

**Published:** 2020-03-16

**Authors:** Nadezhda Vishnevskaya, Vlada Shakhnazarova, Alexander Shaposhnikov, Olga Strunnikova

**Affiliations:** 1All-Russia Research Institute for Agricultural Microbiology, Podbelsky shosse 3, 196608 St Petersburg-Pushkin, Russia; navishnevskaya@rambler.ru (N.V.); shahnazarova-v@mail.ru (V.S.); ai-shaposhnikov@mail.ru (A.S.); 2Department of Agrochemistry, Saint Petersburg State University, 16 Liniya V.O. 29, 199178 St Petersburg, Russia

**Keywords:** plant-microbe interactions, root exudates, chemotropism, colonization, growth stimulation of germ tubes

## Abstract

The aim of this study was to find out why after joint inoculation of the substrate with the phytopathogenic fungus *Fusarium culmorum* and the antagonistic bacterium *Pseudomonas fluorescens* the amount of the fungus on the root surface in the beginning of the colonization was greater on the roots colonized by the bacterium than on control roots. This phenomenon is especially interesting because joint inoculation with *P. fluorescens* was always followed by a considerable decrease in the incidence of Fusarium root rot. In two experiments barley was grown in sterile vermiculite inoculated only with *F. culmorum*, only with *P. fluorescens* and jointly with the fungus and the bacterium. In the control, vermiculite was not inoculated with any microorganisms. After the removal from the vermiculite, barley plants were transferred into deionized water for the collection of root exudates. The duration of barley growth in the vermiculite and in the water was different in the two experiments. The exudates were tested for their ability to elicit chemotropism in *F. culmorum* and influence its growth. We did not observe any chemotropism of *F. culmorum* towards barley root exudates. However, the exudates of the barley colonized by the bacterium stimulated the growth of fungal germ tubes. Using an ultra-performance liquid chromatography system, we found that experimental conditions influenced the quantitative composition of the exudates. The amount of amino acids in the solution of exudates decreased considerably after a prolonged growth of control barley in water, while the presence of *P. fluorescens* resulted in a considerably increase of the amount of amino acids in the exudates. The exudates of barley colonized by *P. fluorescens* contained much more glucose, lactic acid and several amino acids than the exudates of control barley. These components are known to be necessary for the growth of *F. culmorum*. Their presence in the exudates of barley colonized by *P. fluorescens* seems to be the reason of a more active colonization by the fungus of barley roots colonized by the bacterium.

## 1. Introduction

*Fusarium culmorum* (W. G. Smith) Saccardo is a common facultative phytopathogenic fungus affecting numerous crops [1]. In grain crops, it causes seedling blight, root rot, foot rot and Fusarium head blight, which may be expressed at all stages of plant development and result in a considerable loss of yield [2,3,4,5,6,7]. The grain of infected crops accumulates trichothecine toxins hazardous to humans and animals [8,9,10]. In eukaryotic cells, these toxins inhibit the synthesis of proteins, DNA and RNA, change the membrane structure and affect cell division and apoptosis [11,12].

Plant diseases caused by soilborne fungi can be controlled with the use of rhizobacteria. [13,14,15,16]. One such rhizobacterium with biocontrol properties is *Pseudomonas fluorescens* strain 2137. In our earlier studies *P. fluorescens* strain 2137 actively suppressed the development of *F. culmorum* 30 both in the soil without plants and in the barley rhizosphere [17,18]. At the same time, we noted that after joint inoculation of the substrate with *F. culmorum* and *P. fluorescens* the amount of the fungus on the root surface in the beginning of barley colonization was greater than in case of inoculation with the fungus only [18,19,20]. Later the amount of *F. culmorum* on barley roots varied considerably both after inoculation with the fungus only and after inoculation with the fungus and the bacterium together. However, an increased amount of *F. culmorum* on the root surface in the first days of colonization in case of joint inoculation with *P. fluorescens* was always followed by a considerable decrease in the incidence of Fusarium root rot [18,19,20]. 

It is unclear what makes *F. culmorum* actively colonize the roots in the presence of *P. fluorescens*. An answer to this question would elucidate the earliest stages of interactions between the plant, the fungus and the bacterium and establish the role of the plant in these interactions. Information about the causes of the increased amount of the pathogen on the roots in the presence of its antagonist accompanied by a biocontrol effect is an essential step in the study of biocontrol mechanisms. 

A more active colonization could be associated with a chemotropic response of the fungus. While the response of rhizosphere bacteria to plant root exudates has often been shown [21,22,23,24,25,26], that of soil fungi is less well studied. Both tomato roots and root exudates induced a significant chemotropic response in *Fusarium oxysporum* f. sp. *lycopersici* microconidia [27]. Exudates of tomato growing in a split-root system and stressed with wounding or salt stimulated the growth especially well, acting as attractants of the biocontrol fungus *Trichoderma harzianum* T22. At the same time, such exudates did not enhance chemotropism in the phytopathogenic fungus *Fusarium oxysporum* f. sp. *lycopersici*, as compared with exudates of unstressed plants. The authors suggested that some biocontrol microorganisms might have developed an ability to sense a broader range of stress signals than plant pathogens [28]. 

The aim of this study was to find out why the amount of *F. culmorum* on the roots of barley colonized by *P. fluorescens* increases at the early stage of colonization. We hypothesized that the root exudates of barley colonized by the bacterium induced a chemotropic response in *F. culmorum*, possibly because the bacterium affected the composition of the exudates and/or some of its own metabolites were added to the exudates. To check our hypothesis, we tested the chemotropism of *F. culmorum* to the root exudates of control barley and the barley colonized by the bacterium and assessed their effect on fungal growth. In order to find out whether the exudates contained substances with a potential positive effect on *F. culmorum*, we performed a comparative analysis of sugars, organic acids and amino acids in the exudates.

## 2. Results 

### 2.1. The Effect of Experimental Conditions on the Amount of F. culmorum and P. fluorescens in the Roots and Root Rot Incidence in Barley Plants 

The amount of *F. culmorum* on the root surface in the presence of *P. fluorescens* increased very early in the process of fungal colonization. Therefore, for the study of this phenomenon we needed the exudates of young barley plants. We performed two experiments specially designed in such a way as to find out whether the composition and the amount of the exudates depended on growth conditions and the duration of exudation. 

The microorganisms and the barley seeds were added to the vermiculite and incubated in potted cultures for 36 h in experiment 2 and for 67 h in experiment 1. Earlier experiments showed that 36 h was a sufficient time for the fungus and the bacterium inoculated into the vermiculite to establish in the rhizosphere and colonize barley roots. Increasing the time to 67 h allowed the pathogen, the antagonist and the plant to develop close interactions in the rhizosphere, which might have affected the intensity of root colonization and the composition of the root exudates. 

In both experiments plants taken out of the vermiculite for the collection of root exudates were transferred into deionized water and kept there for 24 h in experiment 1 and for 96 h in experiment 2. The amount of the fungus and the bacterium in the roots was determined after barley growth in the vermiculite. The number of plants with root rot symptoms was assessed in the end of the experiments. In experiment 2, we also assessed the amount of the fungus and the bacterium in the solution of exudates and their amount in 132-h-old barley roots in the end of the experiment (Table 1).

*F. culmorum* and *P. fluorescens* inoculated into the vermiculite colonized barley roots in both experiments. In the experiment with inoculation by the fungus only, its amount was greater in the roots growing in the vermiculite for 36 h (experiment 2) than in those growing for 67 h (experiment 1); the number of plants with root rot symptoms was the same in both experiments (Table 1). The amount of *P. fluorescens* in the roots in case of inoculation of the vermiculite with the bacterium only was also higher in 36-h-old barley (experiment 2). In case of joint inoculation of the vermiculite with the fungus and the bacterium, the amount of the fungus in the roots increased considerably while the amount of the bacterium, on the contrary, decreased considerably as compared with the inoculation by the fungus and the bacterium separately. At the same time, the joint inoculation with *F. culmorum* and *P. fluorescens* resulted in a decrease in the number of plants with root rot symptoms in both experiments (Table 1). 

In experiment 2, the amount of the fungus and the bacterium in barley roots decreased after 96 hours of growth in water. However, in this experiment, too, the simultaneous presence of *F. culmorum* and *P. fluorescens* in the roots resulted not only in an increased amount of the fungus but also in a decreased amount of the bacterium. During 96 h of barley growth in the water, *F. culmorum* and *P. fluorescens* were also present in the solution of the exudates. The amount of bacterium in the solution was greater than that of the fungus (Table 1). 

### 2.2. Testing the Chemotropic Response in F. culmorum Towards Barley Root Exudates 

Root exudates collected in the two experiments were tested for their ability to elicit chemotropic response in *F. culmorum*. Microscopic examination showed that approximately the same number of germ tubes were oriented towards wells with exudates and wells with water (Figure 1). We found no differences in the number of germ tubes growing towards exudates of control barley and exudates of roots colonized by *P. fluorescens*. 

### 2.3. Influence of Root Exudates on the Growth of F. culmorum 

In order to find out why *F. culmorum* actively colonized barley roots with *P. fluorescens,* we also assessed the influence of root exudates collected in the two experiments on the fungal growth (Figure 2). Microscopic examination showed that the proportion of germinated macroconidia of *F. culmorum* was the same in the experiments with their incubation in the exudates of control barley, the exudates of barley inoculated with the fungus and the bacterium, in Czapek-Dox broth (CDB) and in water. However, the growth of germ tubes was considerably stimulated after incubation of 

*F. culmorum* macroconidia in the exudates of barley colonized by the bacterium as compared with the exudates of control barley (Figure 2). Incubation in the exudates of control barley and the exudates of barley colonized by the fungus influenced the length of the germ tubes in the same manner. The germ tubes were somewhat shorter in the experiment with incubation of macroconidia in the exudates of the roots colonized jointly by the fungus and the bacterium (Figure 2).

It would seem that these results are easy to explain by the influence of the nutrition available to the fungus, as evidenced by its minimum development in water and maximum development in CDB. However, a greater length of germ tubes in the exudates of barley colonized by the bacterium as compared with the control barley exudates apparently indicates that the exudates of barley colonized by *P. fluorescens* contained substances stimulating fungal growth. 

### 2.4. The Effect of P. fluorescens on the Growth of F. culmorum 

In order to check whether the bacterium itself could stimulate the growth of the fungus, we assessed the influence of *P. fluorescens* on the germination of *F. culmorum* macroconidia. Fungal macroconidia and bacterial cells were suspended in a mineral solution and kept for 6 h to ensure their interaction. In control, fungal macroconidia were incubated in a mineral solution without the bacterium. *P. fluorescens* considerably inhibited the germination of macroconidia (Figure 3A) and suppressed fungal growth (Figure 3B).

### 2.5. Quantitative Composition of Sugars, Organic Acids and Amino Acids in Root Exudates of Control Barley, Barley Colonized by P. fluorescens and Jointly by F. culmorum + P. fluorescens

Our results showed that *P. fluorescens* did not stimulate the growth of *F. culmorum* but, on the contrary, suppressed it. However, root exudates of barley colonized by the bacterium stimulated early stages of fungal growth. We hypothesized that the stimulation might be associated with the presence of nutrients necessary for fungal growth in the root exudates. 

We identified the substances utilized by *F. culmorum* and *P. fluorescens* by comparing the amount of initial and residual components after five days of their growth in the solution of barley root exudates in a previous study [29]. The results are summarized in Table 2 (only the substances utilized most actively by the fungus during its growth are given). We found that the fungus and the bacterium mostly used sugars, especially glucose. *P. fluorescens* utilized organic acids and amino acids more actively than *F. culmorum*. Glucose was the preferable source of carbon for the growth of *F. culmorum*.

In order to reveal the components potentially enhancing the growth of *F. culmorum*, we performed a comparative analysis of the amount of sugars, organic acids and amino acids in the root exudates of barley collected in the two experiments.

The comparative analysis showed that the content of sugars was considerably higher in the exudates of barley colonized by the bacterium than in control barley exudates, mostly because of a considerable increase in the glucose level (Table 3). In exudates of control barley and barley colonized by the bacterium (experiment 2) the content of sugars was lower than in exudates in experiment 1 (Table 3). However, in this experiment, too, the level of glucose in the exudates of barley colonized by the bacterium was statistically significantly higher than in the control barley exudates.

The amount of sugars in the exudates of barley colonized jointly by the fungus and the bacterium collected in both experiments was lower than in the exudates of control barley and barley colonized only by the bacterium. This was due to an active utilization of sugars by both microorganisms in barley roots and in the solution of root exudates. A decreased amount of glucose in the exudates in the presence of *P. fluorescens* noted in experiment 2 could be associated with a decrease in the total amount of sugars as well as by the utilization of glucose by the bacterium itself, which was present in the roots and in the solution of exudates during the 96 h of barley growth in water (Table 1). 

In exudates collected in experiment 1 the total content of organic acids in the exudates of barley colonized by the bacterium was somewhat lower than in those of control barley (Table 4). However, the amount of propionic acid and lactic acid was higher in the exudates of barley colonized by the bacterium than in the control barley exudates. *F. culmorum* did not utilize propionic acid but actively utilized lactic acid (Table 2). The amount of propionic acid also increased in the exudates of barley jointly colonized by the fungus and the bacterium. In the exudates of barley colonized by the bacterium collected in experiment 2 the composition of dominating acids changed: the levels of succinic and, especially, acetic acid increased (Table 4). An increased amount of some organic acids in the exudates of barley roots colonized by *P. fluorescens* may be associated with the metabolism of the bacterium itself. A decreased amount of organic acids in the exudates of barley colonized by jointly the fungus and the bacterium can be explained by the fact that the microorganisms utilized these substances during their growth. 

The amount of amino acids in the exudates of barley collected in experiment 1 was higher in the exudates of control barley than in the exudates of barley colonized by the bacterium (Table 5). Obviously, amino acids as the source of nitrogen were used by *P. fluorescens* for its growth. The exudates of control barley collected in experiment 2 contained a much lesser amount of all amino acids except γ-aminobutyric acid, the amount of which increased. On the contrary, in the exudates of barley colonized by the bacterium the amount of most amino acids increased considerably (Table 5). Some of these amino acids were essential for the growth of *F. culmorum* (Table 2). 

In root exudates of barley colonized by the fungus and the bacterium, the amount of all amino acids decreased dramatically in experiment 2, in which the amino acids from the exudate solution were utilized not only by the bacterium but also by the fungus, and possibly by the plant itself. 

## 3. Discussion 

In our experiments, joint inoculation of the vermiculite with *F. culmorum* and *P. fluorescens* resulted in an increased amount of the fungus on the barley roots as compared with the inoculation of the substrate with the fungus only (Table 1). This phenomenon has also been observed in our previous studies. The elucidation of its causes was the aim of this research. 

We ascertained that an increased amount of *F. culmorum* on the roots was not associated with its chemotropism towards the exudates of barley colonized by *P. fluorescens* (Figure 1). In our experiments, the amount of macroconidia germ tubes oriented towards exudates and towards water was approximately the same. In the present study, we did not register any influence of root exudates and CDB on the germination of *F. culmorum* macroconidia and the chemotropism of germ tubes, contrary to what has been noted in *F. oxysporum* [27]. The reason behind this difference may be that in *F. culmorum* the germination and an initial stage of germ tube growth are less dependent on the exogenous nutrition than in *F. oxysporum.* Large macroconidia of *F. culmorum* can provide the growing hypha with nutrients for a longer time that small microconidia of *F. oxysporum*. However, the growth rate of *F. culmorum* macroconidia germ tubes did depend on the exogenous nutrition. This explains the increased length of the germ tubes after the incubation of macroconidia in the root exudates of barley colonized by the bacterium (Figure 2), which contain many more components necessary for the fungal growth than the control barley exudates (Table 3, Table 4 and Table 5). The dependence of the germination level on the size of the fungal structure has also been shown for *Fusarium oxysporum* f. sp. *lycopersici*. The germination rate of its microconidia in root exudates of tomato differed greatly, and did not exceed 50%–60%, whereas the germination level of chlamydospores was the same in different exudates, reaching 100% [30].

However, the root exudates of barley colonized by the bacterium stimulated the growth of macroconidia germ tubes while the root exudates of control barley did not (Figure 2). It has been shown that the germination of *F. oxysporum* f. sp. *lycopersici* microconidia is stimulated by tomato root exudates [31,32,33]. The root exudates from transgenic insect-resistant cotton significantly promote the spore germination and mycelial growth of cotton *F. oxysporum* [34]. In our experiments, however, the growth of *F. culmorum* macroconidia germ tubes was more strongly stimulated by the root exudates of barley colonized by *P. fluorescens* than by the other root exudates (Figure 2). The greatest stimulation effect was observed during incubation of macroconidia in CDB (Figure 2). CDB contains much more carbohydrates than root exudates and does not contains any substances toxic for the fungus. On the contrary, barley root exudates contain aromatic carboxylic acids (ACA), which have antimicrobial properties. Earlier we showed that ACA suppressed the growth of *F. culmorum* [35]. 

The fact that we did not reveal any chemotropism of *F. culmorum* to exudates indicates that the exudates of barley colonized by the bacterium do not contain any components that can attract the fungus to the roots. However, we observed the stimulation of the fungal growth by root exudates of barley colonized by the bacterium. In the absence of attractants, growth (and thus a more intensive colonization of the roots) may be stimulated by additional nutrition. This was supported by the results of stimulation of the fungal growth during its incubation in the nutrient medium (Figure 2). We analyzed sugars, organic acids and amino acids, because they are known to be essential for the growth of the rhizosphere microorganisms [36,37,38].

Comparative analysis showed that the exudates of barley colonized by the bacterium contained more components necessary for the fungal growth than the control barley exudates (Table 3, Table 4 and Table 5). Our analysis showed that the root exudates of barley colonized by *P. fluorescens* contained the highest amounts of glucose (Table 3), an increased amount of lactic acid (Table 4) and significantly greater amounts of phenylalanine, tryptophan and other amino acids than the control barley exudates (Table 5). It is possible that in this study and in our earlier experiments the exudates of barley colonized by *P. fluorescens* stimulated not only the growth of germ tubes of *F. culmorum* macroconidia but also further growth of its mycelium. This stimulation could lead to a more active colonization by the fungus of the roots colonized by the bacterium than sterile (control) roots. 

Joint inoculation of the vermiculite with the pathogen and its antagonist resulted in a greater amount of *F. culmorum* but a lesser amount of *P. fluorescens* in barley roots as compared to the mono-inoculation (Table 1). We have already demonstrated that *F. culmorum* 30 and *P. fluorescens* 2137 jointly inoculated into the vermiculite successfully colonized the roots and occupied almost the same root zones [19,39]. When both microorganisms were present on the root surface, the amount of each could later fluctuate and, in particular, decrease significantly. However, regardless of the fluctuations of the abundance of *P. fluorescens* on the root surface, its presence always resulted in a decrease in the number of diseased barley plants [18,19,20]. Apparently, the final outcome (the disease incidence in barley) was determined not by the amount of the phytopathogenic fungus and/or antagonistic bacterium in the roots but by their functional activity. Our further studies will focus on the mechanisms of interactions between *F. culmorum*, *P. fluorescens* and barley plants. 

We noted a considerable effect of conditions on the quantitative composition of barley exudates in our experiments. The exudates collected in experiment 2, after 96 h of barley growth in water, contained a lesser amount of sugars, organic acids and especially amino acids than barley exudates collected after 24 h of growth in experiment 1. In experiment 2 the plants, kept in deionized water for a long time, could photosynthesize but were actually starving. Under these conditions they probably used some components of their own root exudates, first of all, amino acids. A recapture of root exudates by the plant is an established fact [40,41,42].

Interestingly, in experiment 2, when the amount of amino acids in the exudates of control barley decreased considerably, it was the presence of *P. fluorescens* in the roots that promoted an increase of their amount in the exudates. It was known that 2,4-diacetylphloroglucinol, commonly produced by rhizosphere pseudomonads, enhances exudation of 16 amino acids from the roots of alfalfa, maize, and wheat [43]. In our study, however, an increased amount of amino acids in the presence of the bacterium was noted only in the exudates collected in experiment 2 (Table 5). In experiment 1, the presence of the bacterium in barley roots even resulted in a decreased amount of amino acids in the exudates as compared to the control. These data indicate that environmental conditions may be a crucial factor influencing the interactions between barley and the biocontrol bacterium *P. fluorescens*. 

The experiment showed that *P. fluorescens* itself had no stimulating effect whatever on *F. culmorum.* It demonstrated an evident antagonistic effect on the fungus (Figure 3). Thus, the stimulation of *F. culmorum* growth by root exudates of barley colonized by the bacterium was associated not with the direct effect of the bacterium itself but with its presence in the roots. Apparently, *P. fluorescens* colonizing the roots influences the composition of root exudates and possibly also adds its own metabolites, which makes the exudates attractive for *F. culmorum*.

## 4. Materials and methods

### 4.1. Study Objects 

We used barley *Hordeum vulgare* L. cv. Belogorsky, which is susceptible to Fusarium root rot, from the Collection of N.I. Vavilov All-Russian Institute of Plant Genetic Resources. Facultative phytopathogenic fungus *Fusarium culmorum* strain 30 isolated from barley roots was grown on Czapek-Dox agar (CDA) for 14 days. Macroconidia were washed off with sterile water, precipitated by centrifuging at 11000 rpm for 15min at 4 °C, resuspended and brought to the required concentration. An antagonistic bacterium *Pseudomonas fluorescens* strain 2137 was grown for 24 h on King’s medium B agar at 28 °C [44]. The cells were suspended in sterile distilled water, precipitated by centrifuging at 10,000 rpm for 15min at 4 °C and resuspended.

### 4.2. Experimental Conditions 

Root exudates of sterile barley and barley colonized by *F. culmorum* and *P. fluorescens* were obtained in two experiments. Both were conducted under sterile conditions in airtight glass pots with a volume of 0.5 L, each containing 24 g of dry vermiculite. In both experiments the vermiculite was inoculated: (1) with a suspension of fungal macroconidia (2 × 10^5^/mL), (2) a suspension of bacterial cells (1.5 × 10^7^), (3) a suspension of fungal and bacterial cells taken in the same concentrations. Vermiculite in the control pots was not inoculated. The total amount of water, including the suspensions of fungal and bacterial cells, was 250 mL per each 100 g of dry vermiculite. Four pots were used in each experiment except the experiment with joint inoculation of the vermiculite with the fungus and the bacterium, when eight pots were used. Barley seeds were sterilized for 30 s in 96% ethanol, washed thrice in sterile water and soaked for 30 min in 1% solution of AgNO_3_. After the removal of the silver nitrate, the seeds were washed once in 1% solution of NaCl and five times in sterile water. The seeds were transferred into sterile Petri dishes with moist filter paper and kept for 48 h. Germinated seeds were transferred into prepared pots with vermiculite, 10 seeds per pot. Barley plants were grown for 67 h in experiment 1 and for 36 h in experiment 2. After that, they were carefully removed from the pots. Excessive vermiculite was removed by washing the roots in sterile water. One plant from each pot was used to determine the amount of the fungus and the bacterium in barley roots by inoculating homogenized roots on CDA and on King’s medium, correspondingly. The remaining plants were transferred into prepared airtight glass pots with sterile deionized water in such a way that the water should cover only the roots. The pots were kept in daylight for 24 h in experiment 1 and for four days in experiment 2. After that, the solutions of root exudates were centrifuged at 11,000 rpm for 15min at 4 °C for precipitation of fungal cells, bacterial cells and root cells. The supernatants were stored at −20 °C until the determination of the composition of root exudates. The roots were examined for the presence of rot symptoms, dried and weighed. 

### 4.3. Testing the Chemotropism of F. culmorum Towards Barley Exudates 

Chemotropism in *F. culmorum* was tested with the use of a technique suggested by Turrà et al. [13] with slight modifications. Macroconidia obtained from a 14-day-old fungal culture grown on CDA were washed off the agar with sterile cold (+4 °C) water. They were centrifuged twice (15 min, 11,000 rpm, +4 °C) and resuspended in cold water in order to remove CDA residues. A suspension of fungal macroconidia (10^5^/mL) in a volume of 5 mL was mixed with 5 mL of warm 1% water agarose (Serva). The mixture was poured into a Petri dish, 8.5 cm in diameter, with an inserted template. Once the agarose solidified, the template made it possible to make in it two parallel wells, 77 × 2 mm in size, located at a distance of 10 mm from each other. One of the wells was filled with 200 µl of the solution of exudates. The other well was filled with 200 µL of sterile water. The Petri dish was incubated at 20 °C. After 13 h of incubation, a line was drawn on the bottom of the dish between the wells, and the gel layer above it was examined under a microscope (Imager A 1, Zeiss Axio, 200 × magnification). The germ tubes growing towards the well with the exudates and those growing towards the well with water were counted. Three Petri dishes were used in each case, and the growth direction of 200 macroconidium germ tubes was assessed in each dish. The chemotropic index was calculated as ((Htest − H*sw*) / Htotal × 100), where Htest is the number of maroconidia germ tubes growing towards the test exudates, Hsw is the number of germ tubes growing towards the sterile water, and Htotal is the total number of germ tubes counted. For each test compound a total of 600 germ tubes were scored. All experiments were performed twice. Statistical analysis was conducted using *t*-test.

### 4.4. Assessment of the influence of barley root exudates on the growth of F. culmorum 

The fungal suspension washed off the medium (2∙10^5^ macroconidia/mL) was mixed with each of the solutions of exudates obtained in two experiments in the proportion of 1:3. For fungal growth control, the suspension of macroconidia was mixed with water (in one case) and CDB (in another case) in the same proportions. Macroconidia were incubated for 6 h at 24 °C. In each case 300 macroconidia of *F. culmorum* were assesed. The proportion of germinated macroconidia to the total number of examined macroconidia and the length of germ tubes were registered. 

### 4.5. Assessing the Effect of P. fluorescens on the Growth of F. culmorum

Fungal macroconidia washed off the medium were diluted with a mineral medium (g/L: Ca(NO_3_)_2_∙ 4H_2_O − 1.18, KNO_3_ − 0.5, KH_2_PO_4_ − 0.136, MgSO_4_∙ 7H_2_O − 0.48) to obtain a concentration of 5∙10^4^ macroconidia/mL. Some of the suspension was left for control. *P. fluorescens* cells washed off the medium were added to the remaining part of the suspension to reach a concentration of 5 × 10^6^ cells/mL. Macroconidia were incubated for 6 h at 24 °C. We microscopically examined 300 macroconidia in each case and registered the proportion of germinated macroconidia from the total number of examined ones. We also measured the length of growth tubes. 

### 4.6. Chromatographic Analysis of Low-Molecular-Weight Components of Root Exudates

The quantitative composition of sugars, organic acids, and amino acids in the collected exudates was evaluated using a Waters ACQUITY UPLC H-Class Ultra Productive Liquid Chromatography (UPLC) system (Waters, USA). The analyses were repeated three times. Solutions of root exudates were filtered under vacuum through 0.45 μm membrane filters and concentrated at 45°C on rotary vacuum evaporator Heidolph Hei-VAP Precision (Heidolph Instruments GMBH & CO KG, Germany) to a volume of 10 mL. Concentrates were passed through a DOWEX 50WX8 100–200 mesh ion exchange resin in hydrogen form (Sigma-Aldrich, Co.) for separation into two fractions: (1) organic acids and sugars, (2) amino acids. The resulting fractions were evaporated to dryness under vacuum and dissolved in 1 mL of Milli-Q water. 

Organic acids were separated in 10mM orthophosphoric acid on a Waters ACQUITY CSH C18 (1.7 μm, 2.1 × 75 mm) column (Waters, CIIIA) at flow rate 0.1 mL/min, column temperature 24 °C and detected on Waters eλPDA detector at a wavelength of 220 nm. 

Sugars were determined by refractometry using a Waters 2414 detector included in the UPLC system. The analysis was performed using the SUPELCOSIL LC-NH2 (5 μm, 4.6 × 250 mm) column (Sigma-Aldrich, Co.) with 75% acetonitrile at flow rate 1 mL/min and column temperature 31 °C. 

To determine the composition of proteinogenic amino acids except L-tryptophan, we used fluorescent derivatives of amino acids obtained and analyzed according to the protocol for the Waters AccQ-Tag method (Waters, USA). L-tryptophan was analyzed without derivatization by separation on Waters ACQUITY UPLC BEH RP18 Shield (1.7 μm, 2.1 × 50 mm) column (Waters, CIIIA) in mixture of 0.1% formic acid (A) and acetonitrile supplied with 0.1% formic acid (B) with gradient at flow rate 0.3 mL/min from 1 to 18% B for 5 minutes, followed by washing with 80% B for 2 minutes and conditioning the column for 3 minutes at 1% B. L-tryptophan was detected with fluorescence detector (λ_ex_ = 280 nm, λ_em_ = 350 nm). 

### 4.7. Statistical Data Analysis 

One-way ANOVA and Student’s t-test were used to compare the number of germ tubes (n = 600) with a certain orientation; the length of germ tubes (n = 300); the quantitative composition of sugars, organic acids and amino acids in root exudates of control barley, barley colonized by the bacterium (n = 4) and barley colonized jointly by the fungus and the bacterium (n = 8). Values given in tables are means ± standard deviations. Error bars in diagrams show standard deviation. Significant differences in germination of macroconidia were analyzed using the Mann–Whitney U-test. Values with different letters are significantly different at *p* ≤ 0.05 when applying Fisher’s LSD test and at *p* ≤ 0.01 when applying Mann–Whitney U-test.

## Figures and Tables

**Figure 1 plants-09-00366-f001:**
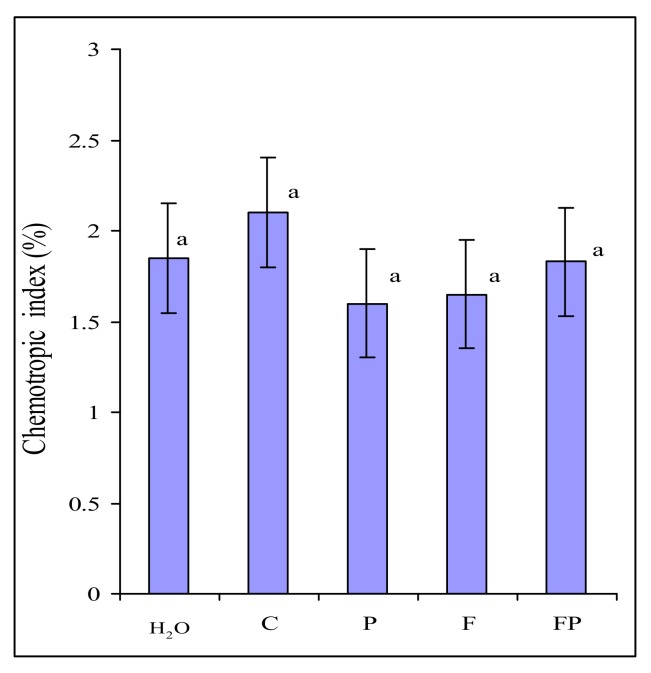
Directed growth of *Fusarium culmorum* macroconidia germ tubes to water (H_2_O) and to exudates of: control barley (C), barley colonized by *Pseudomonas fluorescens* (P), barley colonized by *F. culmorum* (F), barley colonized by both *P. fluorescens* and *F. culmorum* (FP). Data are presented as the mean from two experiments. *n* = 600 germ tubes. Error bars show standard deviation. Values of different letters are significantly different at *p* ≤ 0.05 when applying Fisher’s LSD test.

**Figure 2 plants-09-00366-f002:**
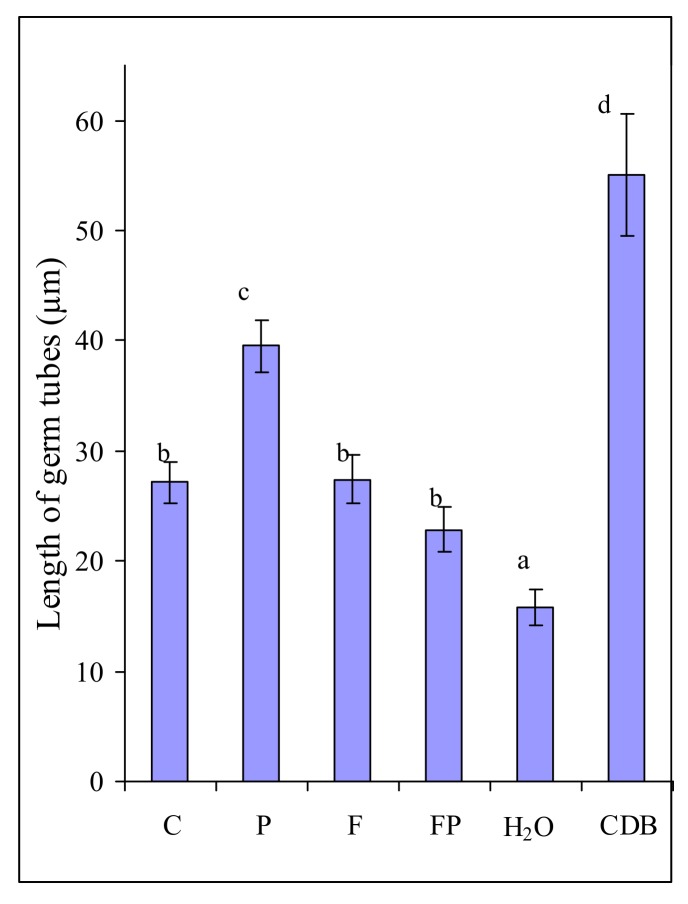
Length of germ tubes of *Fusarium culmorum* macroconidia after 6 h of incubation in solutions of exudates of control barley (C), barley colonized by *P. fluorescens* (P), barley colonized by *F. culmorum* (F) and barley colonized by both *P. fluorescens* and *F. culmorum* (FP), in sterile water (H_2_O), in liquid medium (CDB). Data are presented as the mean from two experiments. *n* = 300 macroconidia. Error bars show standard deviation. Values of different letters are significantly different at *p* ≤ 0.05 when applying Fisher’s LSD test.

**Figure 3 plants-09-00366-f003:**
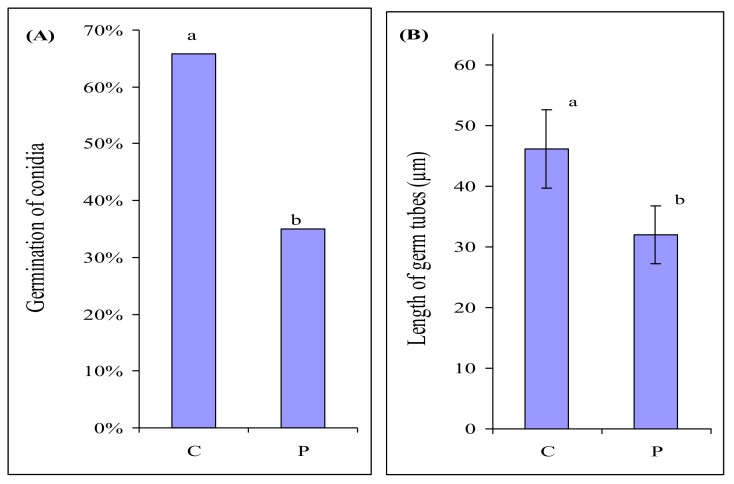
Germination of *Fusarium culmorum* macroconidia (**A**), length of germ tubes of *F. culmorum* macroconidia (**B**) after 6 h of incubation in the mineral medium in the absence (C) and presence of *P. fluorescens* (P). *n* = 300 macroconidia. Significant differences in germination of macroconidia were analyzed using Mann–Whitney U-test. Error bars show standard deviation. Values of different letters are significantly different at *p* ≤ 0.05 when applying Fisher’s LSD test, and at *p* ≤ 0.01 when applying Mann–Whitney U-test.

**Table 1 plants-09-00366-t001:** Barley disease incidence and the amount of *Fusarium culmorum* and *Pseudomonas fluorescens* in the roots and in water.

Conditions of the Experiment	Vermiculite Inoculation	Diseased Plants (%)	Amount of the Fungus after the Growth of Barley			Amount of the Bacterium after the Growth of Barley		
			in the Vermiculite CFU/g of Root (10^3^)	In Water		In the Vermiculite CFU/g of Root (10^5^)	In Water	
				CFU/g of Root (10^3^)	CFU/mL (10^2^)		CFU/g of Root (10^5^)	CFU/mL (10^3^)
**Experiment 1:** The growth of barley in the vermiculite ‒ 67 h, the exudation in water ‒ 24 h. The age of barley at the end of the experiment - 91 h	None	0						
	*F. culmorum*	16	1.2 ± 0.3 a					
	*P. fluorescens*	0				95.7 ± 5 a		
	*F. culmorum + P. fluorescens*	8	29 ± 10 b			48.7 ± 14 b		
**Experiment 2:** The growth of barley in the vermiculite ‒ 36 h, the exudation in water ‒ 96 h. The age of barley at the end of the experiment -132 h	None	0						
	*F. culmorum*	16	19.9 ± 3 a	11.7 ± 3.6 c	0.5 ± 0.05 a			
	*P. fluorescens*	0				221.9 ± 63 a	83.3 ± 25 c	15 ± 2 a
	*F. culmorum + P. fluorescens*	10	158.9 ± 35 b	40.3 ± 6 d	3,3 ± 1 b	90,7 ± 27 c	46.7 ± 11 b	12,3 ± 1,8 a

Values in the same experiment with different letters are significantly different at *p* ≤ 0.05 when applying Fisher’s LSD test. Values are means ± standard deviations.

**Table 2 plants-09-00366-t002:** Components of barley root exudates consumed by *Fusarium culmorum* and *Pseudomonas fluorescens* during 5 days of growth in the exudate solution. (Based on the results of earlier studies [29]).

Components of Barley Root Exudates	Initial Amount (µg/mL)	Amount (µg/mL) Consumed by	
		*F. culmorum*	*P. fluorescens*
**Sugars:**			
**glucose**	357.6 ± 68	328.8 ± 59 a	240 ± 23.5 b
**fructose**	34.7 ± 7.3	16.8 ± 3.8 a	24.5 ± 4.7 a
**other**	147.1 ± 20.7	1.1 ± 0.2 a	1 ± 0.1 a
**Organic acids:**			
malic	11.7 ± 2.1	6 ± 1.3 a	9.2 ± 2 a
lactic	9 ± 1.6	8 ± 1.9 a	7.7 ± 1.4 a
other	14.8 ± 2.8	0.1 ± 0.01a	11 ± 3.3 b
**Amino acids:**			
proline	1.75 ± 0.22	0.9 ± 0.2 a	0.4 ± 0.06 b
phenylalanine	1.2 ± 0.2	0.8 ± 0.1 a	1.1 ± 0.2 b
tryptophan	0.85 ± 0.2	0.75 ± 0.08 a	0.8 ± 0.09 b
histidine	0.55 ± 0.13	0.33 ± 0.09 a	0.47 ± 0.14 a
tyrosine	0.45 ± 0.11	0.32 ± 0.1 a	0.4 ± 0.11 a
valine	0.57 ± 0.12	0.39 ± 0.1 a	0.5 ± 0.1 a
lysine	0.35 ± 0.1	0.15 ± 0.05 a	0.32 ± 0.07 b
leucine	0.69 ± 0.17	0.35 ± 0.09 a	0.67 ± 0.12 b
isoleucine	0.42 ± 0.13	0.19 ± 0.04 a	0.4 ± 0.08 b
other	2.08 ± 0.25	0.35 ± 0.1 a	1.47 ± 0.2 b

Values with different letters are significantly different at *p* ≤ 0.05 when applying Fisher’s LSD test. Values are means ± standard deviations.

**Table 3 plants-09-00366-t003:** Amount of sugars in the exudates of control barley colonized by *P. fluorescens* and jointly by *F. culmorum + P. fluorescens* collected in the two experiments.

Sugars	Amount of Sugars (μg/ g DW) in the Absence and Presence of Microbes					
	Experiment 1			Experiment 2		
	None	*P. fluorescens*	*F. culmorum + P. fluorescens*	None	*P. fluorescens*	*F. culmorum + P. fluorescens*
Melibiose	62 ± 14 a	29.5 ± 15 b	22 ± 9 b	ND	ND	ND
Maltose	2968 ± 590 a	3500 ± 350 a	492 ± 190 b	736 ± 244 a	658 ± 151 a	286 ± 83 b
Sucrose	21 ± 10 a	60 ± 30 ab	9.5 ± 5 ac	62 ± 29 a	2 ± 1.1 b	2.7 ± 1.2 b
**Glucose**	1800 ± 485 a	**4500 ± 845 b**	1346 ± 497 a	122.5 ± 27 a	**490 ± 152 b**	67 ± 31 a
Fructose	140 ± 30 a	210 ± 100 a	133 ± 40 a	132 ± 37 a	32 ± 23 b	21 ± 6 b
Arabinose	729 ± 195 a	359 ± 39 b	104 ± 28 c	141 ± 43 a	167 ± 79 a	89 ± 41 a
Xylose	51 ± 9 a	161 ± 27 b	84 ± 28 a	31 ± 9 a	81 ± 19 b	13 ± 7 c
Ribose	199 ± 49 a	127 ± 14 b	150 ± 48 ab	144 ± 58 a	133 ± 72 a	87 ± 37 a

Values in the same experiment with different letters are significantly different at *p* ≤ 0.05 when applying Fisher’s LSD test. Values are means ± standard deviations. ND = not detected. Substances most actively utilized by *F. culmorum* are marked in bold.

**Table 4 plants-09-00366-t004:** Amount of organic acids in the exudates of control barley colonized by *P. fluorescens* and of barley jointly colonized by *F. culmorum* and *P. fluorescens* collected in the two experiments.

Organic Acids	Amount of Organic Acids (μg/ g DW) in the Absence and Presence of Microbes					
	Experiment 1			Experiment 2		
	None	*P. fluorescens*	*F. culmorum + P. fluorescens*	None	*P. fluorescens*	*F. culmorum + P. fluorescens*
Pyroglutamic	12 ± 2 a	4 ± 2 b	0.7 ± 0.3 c	18 ± 1	ND	ND
Propionic	523 ± 46 a	2100 ± 298 b	1980 ± 280 b	74 ± 36 a	ND	188 ± 86 a
Fumaric	11 ± 5 a	1.4 ± 0.5 b	2.5 ± 0.6 b	0.7 ± 0.3 a	0.3 ± 0.13 a	ND
Acetic	1673 ± 344 a	912 ± 45 b	919 ± 108 b	1654 ± 334 a	7865 ± 1320 b	2184 ± 832 a
**Lactic**	284 ± 76 a	**579 ± 128 b**	340 ± 78 a	196 ± 10 a	45 ± 8 b	37 ± 14 b
Succinic	935 ± 229 a	60 ± 19 b	132 ± 52 c	1363 ± 341 a	1173 ± 296 a	295 ± 91 b
t-Aconitic	15 ± 3 a	10 ± 1 b	6 ± 0.8 c	210 ± 7 a	49 ± 18 b	2.6 ± 1.3 c
Malic	1400 ± 298 a	290 ± 82 b	312 ± 97 b	198 ± 21 a	120 ± 6 b	ND
Pyruvic	169 ± 33 a	81 ± 15 b	52 ± 11 c	68 ± 4.5 a	79 ± 13 a	33 ± 6 b
Citric	25 ± 3 a	6 ± 0.8 b	6 ± 0.8 b	309 ± 93 a	166 ± 48 b	2.2 ± 0.5 c
Oxalic	65 ± 14 a	38 ± 8 ab	25 ± 7 b	ND	ND	ND

Values in the same experiment with different letters are significantly different at *p* ≤ 0.05 when applying Fisher’s LSD test. Values are means ± standard deviations. ND = not detected. Substances most actively utilized by *F. culmorum* are marked in bold.

**Table 5 plants-09-00366-t005:** The amount of amino acids in exudates of control barley, barley colonized by *P. fluorescens* and barley jointly colonized by *F. culmorum* and *P. fluorescens* collected in the two experiments.

Amino acids	Amount of amino acids (μg/ g DW) in the absence and presence of microbes					
	Experiment 1			Experiment 2		
	None	*P. fluorescens*	*F. culmorum+ P. fluorescens*	None	*P. fluorescens*	*F. culmorum + P. fluorescens*
Phenylalanine	36 ± 1.8 a	21 ± 2.6 b	4 ± 0.5 c	2.8 ± 0.2 a	17 ± 1.3 b	5.4 ± 0.1 c
**Leucine**	42 ± 4.5 a	17 ± 1.8 b	10 ± 2 c	7.5 ± 0.2 a	**45 ± 2 b**	6 ± 0.2 a
**Isoleucine**	27 ± 1 a	10 ± 1.3 b	7 ± 0.8 b	1.9 ± 0.1 a	**34 ± 3 b**	1.9 ± 0.1 a
**Lysine**	25 ± 5 a	12 ± 1.6 b	9 ± 1.1 b	8 ± 0.2 a	**50 ± 2.5 b**	4 ± 0.1 c
Ornitine	3 ± 0.9 a	0.8 ± 0.2 b	0.6 ± 0.1 b	0.25 ± 0.1 a	0.5 ± 0.15 a	0.6 ± 0.15 a
Methionine	4.3 ± 0.9 a	1.7 ± 0.6 b	0.7 ± 0.2 c	0.9 ± 0.1 a	4.5 ± 0.38 b	1 ± 0.1 a
**Valine**	38 ± 6 a	14 ± 2.6 b	7 ± 1.3 c	8 ± 0.3 a	**70 ± 3.5 b**	3 ± 0.2 c
**Tyrosine**	23 ± 1.5 a	10 ± 1.1 b	3.2 ± 0.5 c	2.5 ± 0.2 a	**23 ± 1.1 b**	1.2 ± 0.1 c
Cysteine	3 ± 0.3 a	6.5 ± 0.9 b	1.2 ± 0.5 a	4 ± 0.2.3 a	0.5 ± 0.1 b	4 ± 1.9 a
γ-Aminobutyric acid	75 ± 8 a	7.2 ± 1.4 b	7.6 ± 1.5 b	290 ± 45 a	130 ± 21 b	2 ± 0.1 c
Proline	129 ± 18 a	43 ± 9 b	16.5 ± 3.5 c	44 ± 7 a	8.5 ± 0.4 b	6 ± 0.2 c
Alanine	41 ± 11 a	29 ± 7 a	28 ± 5 a	20 ± 1 a	23 ± 1 a	7.2 ± 0.4 b
Threonine	7.2 ± 0.8 a	3.5 ± 0.6 b	3.5 ± 0.5 b	0.8 ± 0.2 a	2.4 ± 0.8 a	0.8 ± 0.2 a
Arginine	14 ± 2.8 a	7.3 ± 0.6 b	0.4 ± 0.1 c	3 ± 0.3 a	8 ± 0.5 b	8.5 ± 0.4 b
**Histidine**	6 ± 0.9 a	5.4 ± 0.9 a	3.9 ± 0.6 a	19 ± 0.8 a	**24.5 ± 1 b**	3.6 ± 0.4 c
Glycine	20 ± 4.5 a	16 ± 3.8 a	9.5 ± 2 c	1.5 ± 0.1 a	21 ± 2 b	3 ± 0.3 c
Glutamic acid	130 ± 24 a	64 ± 7 b	82 ± 12 c	4 ± 0.25 a	12 ± 1.1 b	5 ± 0.4 a
Serine	45 ± 11 a	19 ± 4 b	12 ± 3 b	22 ± 1.5 a	84 ± 3 b	3 ± 0.2 c
Aspartic acid	66 ± 11 a	15 ± 3 b	39 ± 9 c	5 ± 0.2 a	39 ± 2.5 b	3 ± 0.1 c
**Tryptophane**	20 ± 6 a	19.5 ± 5 a	2.4 ± 0.7 b	0.7 ± 0.1 a	**9 ± 0.4 b**	3 ± 0.2 c

Values in the same experiment with different letters are significantly different at *p* ≤ 0.05 when applying Fisher’s LSD test. Values are means ± standard deviations. Substances most actively utilized by *F. culmorum* are marked in bold.

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
