# Peer review of "The Role of Root Exudates of Barley Colonized by *Pseudomonas fluorescens* in Enhancing Root Colonization by *Fusarium culmorum"

_plants, 2020, doi:10.3390/plants9030366_

Round 1

Reviewer 1 Report

The revised version of the manuscript by Vishnevskaya  and co-workers has been reviewed. The authors have satisfactorily addressed all the comments raised by this reviewer and the manuscript has considerably improved. I have only minor comments on this manuscript:

  • Rephrase the following sentence (page 4): “...culmorum and P. fluorescens colonized the solution of the exudates during 96 h of barley growth in the water, the bacterium in a larger amount than the fungus (Table 1).”
  • In general, full sentences should not be placed in parentheses in the middle of a paragraph (e.g. “...experiment 1. (Earlier experiments showed that 36 h was a sufficient time for the fungus and the bacterium inoculated into the vermiculite to establish in the rhizosphere and colonize barley roots.) Increasing.....”).
  • Table 2 is based on previous published results and should be moved to supporting information.
  • Page 6: A parenthesis is needed at the end of “(Based on the results of earlier studies [29]”.
  • The authors should carefully check the numbering of figures and tables throughout the manuscript. For example: (i) In page 7. “... mostly because of a considerable increase in the glucose level (Fig. 3, А)”. The provided Fig. 3A does not give information about glucose content levels in root exudates; (ii) In page 7: “This was due to an active utilization of sugars by both microorganisms in barley roots and in the solution of root exudates (Table 1).” I guess the authors are referring to Table 3 since Table 1 provides information about the barley disease incidence and fungal and bacterial levels in roots and water; (iii) In page 9: “...of the substrate with the fungus only (Table).” Which table are the authors referring to?
  • Page 10: lycopersici” should be in italics.

Author Response

We are very grateful to the reviewers for their attention to the revised manuscript and valuable comments.

We thank Reviewer 1 for the careful analysis of our manuscript and critical remarks.

The revised version of the manuscript by Vishnevskaya and co-workers has been reviewed. The authors have satisfactorily addressed all the comments raised by this reviewer and the manuscript has considerably improved. I have only minor comments on this manuscript:

Rephrase the following sentence (page 4): “...F.culmorum and P. fluorescens colonized the solution of the exudates during 96 h of barley growth in the water, the bacterium in a larger amount than the fungus (Table 1).” The sentence has been rephrased.

  • In general, full sentences should not be placed in parentheses in the middle of a paragraph (e.g. “...experiment 1. (Earlier experiments showed that 36 h was a sufficient time for the fungus and the bacterium inoculated into the vermiculite to establish in the rhizosphere and colonize barley roots.) Increasing.....”). Thank you for the comment. We have removed the parenthesis.
  • Table 2 is based on previous published results and should be moved to supporting information.

Table 2 has been moved to supporting information.

  • Page 6: A parenthesis is needed at the end of “(Based on the results of earlier studies [29]”. Thank you, a parenthesis is now in place.
  • The authors should carefully check the numbering of figures and tables throughout the manuscript. For example: (i) In page 7. “... mostly because of a considerable increase in the glucose level (Fig. 3, А)”. The provided Fig. 3A does not give information about glucose content levels in root exudates; - Thank you, the numbering has been corrected. We apologize for this mistake, the erroneous reference slipped in from the first version of the manuscript.
  • (ii) In page 7: “This was due to an active utilization of sugars by both microorganisms in barley roots and in the solution of root exudates (Table 1).” I guess the authors are referring to Table 3 since Table 1 provides information about the barley disease incidence and fungal and bacterial levels in roots and water; - This was actually a reference to Table 1 but its purpose was merely to confirm that both the fungus and the bacterium were present in barley roots and in the solution of root exudates. To avoid confusion, we have removed this reference altogether.
  • (iii) In page 9: “...of the substrate with the fungus only (Table).” Which table are the authors referring to?

- This was a reference to Table 1. The correct table number has been inserted. We apologize for this mistake.

  • Page 10: lycopersici” should be in italics. – Thank you, the word has been italicized.

Reviewer 2 Report

I am satisfied with the answers of the authors and the refinement of the manuscript.

I believe that the manuscript can be accepted for publication.

Author Response

We are very grateful to the reviewer 2 for valuble comments of our manuscript.

We thank Reviewer 2 for a careful consideration of our manuscript and critical remarks.

This manuscript is a resubmission of an earlier submission. The following is a list of the peer review reports and author responses from that submission.

Round 1

Reviewer 1 Report

The article by Vishnevskaya et al entitled "The role of root exudates of barley by Pseudomonas fluorescens in enhancing root colonization by Fusarium culmorum" deals with tripartite interaction between plant roots, bacteria and a phytopathogenic fungus. Interestingly, presence of bacteria modified interaction between plant roots and the studied fungus. Observed stimulation of fungal colonization on the root but decreased root rot symptoms raised questions about the nature of the effectors. Root exudates have been studied in absence and presence of the bacteria and might explain the observed stimulation of fungal growth. Objectives of the study are clearly given in the end of the Introduction section. This study sounds interesting and adds light on questions about early interactions between plants and microorganisms.

Major points:

(1) Results: Please give some more explanations in the beginning of the Results section. The first paragraph appears very confusing without explaining in one or two phrases the mentioned experiments. For example, in the first phrase is written "in both experiments", but the reader has no information about these experiments before. The second phrase starts with "inoculation by the fungus only", but the setup of the experiments is still not clear. Moreover in the third phrase, it is the amount of the bacteria "in case of mono-inoculation"? All this paragraph, that is really not clear refers to the Table, but nomenclature between the text and the Table is contradictory!?

(2) Results/Table: There are several issues with the Table that need to be corrected. First, "Experiment 1" and "Experiment 2" appears to be different in the text and in the Table!? In the text (first paragraph in results, but also Materials and methods section, page 10), experiment 2 refers to 36 h, experiment 1 to 67 h. But, in the Table, Experiment 1 is indicated as "One day" and Experiment 2 as "Four days"? So, both indications are in contradiction and also, in addition, different as e.g. 67 h are not equal to four days (throughout the manuscript!).

(3) Results/Table: Amount of bacterium is not given for treatment with bacteria only for Experiment 1? The statement (lines 72-74) can not be found with the values given in the Table. The value for "root rot symptoms" in Experiment 2 is surprisingly found in the line with the bacteria-only treatment?

(4) Results: To understand the results in Figure 2 representing fungal growth stimulation by root exudates from roots colonized by bacteria alone compared to either control roots or to combined colonization by fungus and bacteria, the authors should analyze these different root exudates. So far, the authors suggest correlation of fungal growth with exudate composition only in comparison between control roots "C" and roots incubated with bacteria alone "P". The third condition "FP" is clearly missing for the exudate analyses.

Minor points:

(5) The amount of fungus and bacteria was really analyzed "in" the roots or "on" the roots?? (title line 68)

(6) What might be the explanation of the decrease of sugars in the exudates in time (4 days less than 1 day)? This result (Figure 3) appears really surprising, also because the setup of the analyses is not clear. Please explain.

(7) Discussion: The interesting point of this study, the contrasting effect of the the presence of the bacteria on fungal growth on one hand but on fungal disease on the other hand should be better underlined and discussed .

(8) English editing: Few mistakes need to be corrected.

Abstract: lines 21-24, please complete as phrase; line 31, than instead of "that".

Introduction: line 42, substrate; line 44, add comma "Later,..."; correct species in italic (lines 47, 48), line 49, colonizing.

Results: correct "one-day-old" (lines 119, 122, 130; page 8 top); line 134, "colonized by the bacterium".

Reviewer 2 Report

In previous studies, the authors of this work observed that the rhizospheric bacterium Pseudomonas fluorescens 2137 promotes barley root colonization by the phytopathogenic fungus, Fusarium culmorum, during initial stages of the colonization process. However, this enhanced root colonization did not result in an increased infection since the strain 2137 acts as a Fusarium biocontrol agent during the colonization of barley rhizosphere. In this follow-up study, Vishnevskaya and co-workers investigate the reasons of the increased rhizosphere colonization by F. colmorum when P. fluorescens is present. They found that the rhizospheric bacterium alters root exudates composition and the authors hypothesise that the increased concentration in some of the components of the exudates is responsible for the enhanced root colonization by the phytopathogenic fungus.

Major comments:

Abstract section should be re-written in a way that better contextualizes the study and its objectives. The abstract section should not include subheadings. The introduction section is excessively brief and mainly presents previous data from the authors without contextualising the current state of knowledge in the research field. For example, the introduction section should include more information about the relevance of chemotaxis for microbial root colonization, crop losses and economical consequences of infections caused by fungi of the Fusarium genus, strategies for the biocontrol of Fusarium, etc. English style of the manuscript needs to be improved. Importantly, the results section is currently written as a compendium of disjointed results that make it difficult to follow the content of the article. The authors should explain why a particular experiment was conducted. In section 2.1. and in figure legend of Figs. 3-5 the authors should better describe the differences between “experiment 1” and “experiment 2”. For example, why were different experimental conditions chosen? Which information were the authors trying to obtain from each of the experimental conditions tested? Statistical analyses need to be added to Figures 1 and 2. Also, internal controls are missing in these figures. For example, in addition to the analysis of the effect of root exudates of barley plant colonized by P. fluorescens, the authors should have investigated the effect of P. fluorescens itself on the length of F. culmorum germ tubes. In other words, are there any signalling molecules produced by P. fluorescens that may be stimulating fungal growth? Lines 117-118: “....special attention was paid to the components necessary for the growth of F. culmorum”. Although some of the compounds that can be used as nutrient source by F. culmorum are described in the Discussion section, to better follow the content of the manuscript, the authors should also specify these compounds in the results section of the manuscript. The authors analyzed and compared the composition in sugars, organic acids and amino acids that are present in barley root exudates in the absence and presence of P. fluorescens. The authors observed considerable differences, some of which are surprising. Thus, bacteria of the genus Pseudomonas are metabolically very versatile and are able to use as nutritional source a wide diversity of amino acids, organic acids and sugars. In addition, they are also capable of producing and releasing into the environment, for example, a great diversity of organic acids. These variables seem not to have been taken into account by the authors when interpreting and evaluating the results obtained. Figures 3-5 should be merged and the data presented in table format.

Specific comments:

Line 30-31: “...colonized by P. fluorescens contained more components necessary for...”. Please, describe these components. Lines 44, 45, 47, 48, etc.: Fusarium culmorum and Pseudomonas fluorescens must be in italics. Although there is only one Table included in this manuscript, it needs to be numbered. In the Table located between pages 2 and 3, is it strictly correct to use “colony forming units (CFU)” to quantify filamentous fungal pupulations? In this table, two different values are given when quantifying the fungal and bacterial populations in roots but the authors do not specify what these values are referring to. I suppose that these values correspond to microbial populations measured at different time points during the course of the experiment but this needs to be specified in the legend of the table.

Reviewer 3 Report

The manuscript is relevant and concerns an important problem of mechanisms of interactions between organisms. However, it requires some corrections and clarifications.

I have several questions:

1, Lines 38-42: The first lines of the Introduction duplicate the 2019 article of these authors (19. Shaposhnikov, A.I.; Vishnevskaya, N.A.; Shakhnazarova, V.Yu.; Belimov, A.A.; Strunnikova, O.K. The role of barley root exudates as nutrition source in the interactions between Fusarium culmorum and Pseudomonas fluorescens. Mycol. Phytopathol. 2019, 53, 301–308 (rus). 1-2, Lines 38-66: In the Introduction, own data of authors are detailed, but the data of other authors cited on chemotaxis and attracting substances are practically not disclosed. 2, Lines 62-66: The aims are formulated as a series of methodological tasks. 2, Line 69-72 and Table: The first column of the table does not match the text “In the experiment with inoculation by the fungus only, its amount was greater in the roots growing in the vermiculite for 36 h (experiment 2) than in those growing for 67 h (experiment 1); the number of plants with root rot symptoms was the same (Table)”. The table shows the other duration of the experiments. 10, line 3 from the top:On the cultivation of plants in vermiculite: «Barley plants were grown for 67 h in experiment 1 and for 36 h in experiment 2».

Next about getting root exudates, P. 10, Lines 7-9 from the top «Other plants were transferred into prepared airtight glass pots with sterile deionized water in such a way that the water should cover only the roots. The pots were kept in daylight for 24 h in experiment 1 and for four days in experiment 2».

Is this a continuation of the above experiments 1 and 2? What was a reason for such duration of cultivation of plant and collection of root exudates?

2: In which root exudates were the macroconidia incubated: after experiment 1 (1 day) or experiment 2 (4 days)? 3 A, B: How can you explain that in four-day exudates of both control barley and barley colonized by bacteria, the sugar content was lower than in one-day exudates? In the Discussion, the authors associate the revealed increase in the amount of fungus on the roots of barley during the joint inoculation of vermiculite F. culmorum and P. fluorescens with competition for nutrients and an increase in the number of individual sugars and amino acids in root exudates in the presence of bacteria. However, for some reason, it does not discuss the possible involvement of other components of root exudates, such as flavonoids, phytoalexins, plant hormones and other signaling molecules involved in interactions of plants with pathogens and PGPR.

I believe that the manuscript can be published after the revision.